# A Visual Tool for Interactively Privacy Analysis and Preservation on Order-Dynamic Tabular Data

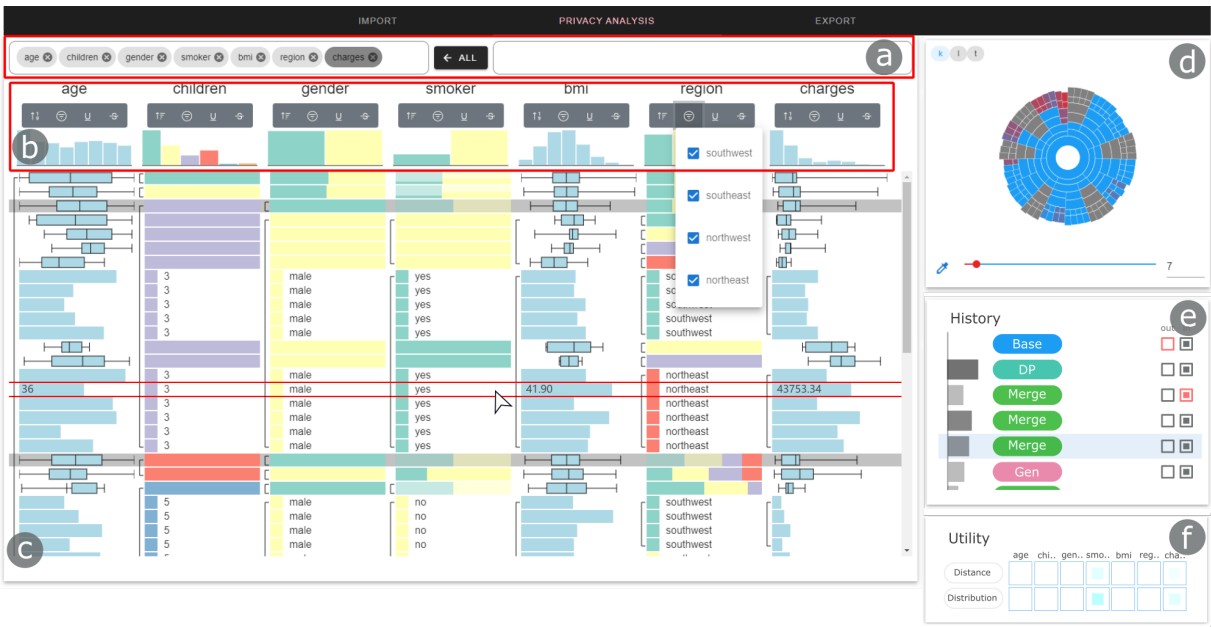

Figure 1: Tabular Privacy Assistant (TPA), a visual tool for the risk analysis and privacy preservation of tabular data with dynamic attribute order. (a) A widget that allows personalized attribute order setting and dynamic adjustment. (b) Statistics of different attributes for overall distribution analysis. (c) The main view for tabular data presentation (box plot means abstract of several items) and interactive privacy enhancement (e.g., choosing five items to merge). (d) Privacy risk tree under the current attribute order (red: privacy breach items on K-anonymity). (e) Historical privacy enhancement operations (allowing backtrack and comparison). (f) Data utility dynamics during interactions.

## ABSTRACT

The practice of releasing individual data, usually in tabular form, is obligated to prevent privacy leakage. With rendered privacy risks, visualization techniques have greatly prompted the user-friendly data sanitization process. Yet, we point out, for the first time, the attribute order (i.e., schema) of tabular data inherently determines the risk situation and the output utility, while is ignored in previous efforts. To mitigate this gap, this work proposes the design and pipeline of a visual tool (TPA) for nuanced privacy analysis and preservation on order-dynamic tabular data. By adapting data cube structure as the flexible backbone, TPA manages to support real-time risk analysis in response to attribute order adjustment. Novel visual designs, i.e., data abstract, risk tree, integrated privacy enhancement, are developed to explore data correlations and acquire privacy awareness. We demonstrate TPA's effectiveness with two case studies on the prototype and qualitatively discuss the pros and cons with domain experts for future improvement.

**Index Terms:** Human-centered computing—Visualization—Visualization techniques—Treemaps; Human-centered computing—Visualization—Visualization design and evaluation methods

## 1 INTRODUCTION

We are all providers and beneficiaries of the collection and release of individual data. Generally maintained as multi-attribute tables, the collected data can be used in various learning, statistic, and decision-making tasks (e.g., disease diagnosing, product recommendation). Alongside the well-known benefits, privacy issues in the publish of data have raised massive concerns recently, as more and more real-world safety violation caused by data leakage and abuse are witnessed [9, 34] and the promulgation of regulations (e.g., GDPR).

The privacy risk stems from the fact that individual identity, although usually anonymized, is correlated and may be re-identified by the other seemingly harmless attributes. As a result, it is obligated for data holders (e.g., organizations, companies) to properly sanitize data before releasing it. Research communities have responded to this critical requirement with many privacy protection techniques, including anonymity [21], differential privacy [10], and synthetic data mixture [1, 42]. With such technical basis, visualization has been introduced recently to facilitate illustrative, understandable, and easy-to-use privacy analysis tools on behalf of the users [6, 7, 37–39, 41]. For example, in [39], visual presentations on privacy exposure level and utility preservation degree are provided for detecting and mitigating privacy issues in tabular data.

Previous visual methods for privacy analysis build on the setting of fixed attribute order, i.e., the target table has fixed columns. However, we find that **the currently unexplored attribute order (i.e., schema) inherently determines the privacy risk situation**

**and the output utility** (Detail analysis in § 3.2). For example, in the process of checking K-anonymity privacy constraints [36] on a sheet, whereas we may find privacy breach on the 3rd attribute and have 5 values changed during protection in an order of 'Age, Work, Disease', we would face a totally different (thornier) privacy context, like privacy breach on the 1st attribute with 10 values changed, in order 'Work, Disease, Age'. As a result, randomly choosing an attribute order, as the existing proposals do, may unfortunately lead to over-protection and unnecessary utility losses.

We are thus motivated to design a flexible visual tool (TPA) that can support and explore order adjustment for nuanced (user-specific, reactive) privacy investigation. The most challenging part for dynamic order is that one should dynamically re-perform risk analysis (e.g., equivalent class parsing) according to the new attribute order. This can be a disaster for existing implementations as it involves aggregation calculations for all combinations under additive prefixes, especially when the sheet owns vast amounts of data items and lots of attributes, indicating significant interaction latency. As a remedy, we adapt the data cube structure with flexibly pre-aggregation to organize the table and use an operation tree to handle order adjustment in real-time (§ 4.1). Additionally, we present a data abstract function for statistically analyzing attribute correlation (§ 5.3) and provide fine-grained utility quantification that estimates the differential impact of each privacy-preserving operation (§ 4.2).

Combined with various privacy enhancement technologies, TPA guides data holders on the risks in their data, and prompt utility losses of preserving operations. The main contributions are as follows:

- We identify the impact of tabular attribute order on privacy analysis, utility loss, and processing costs. We propose a new tool to explore such a property by adopting data cube to guarantee real-time interaction.

- We leverage multi-dimensional value distance to measure utility change at the back-end. We use abstract extraction for inter-attribute relationship analysis and design an intuitive risk tree that semantically bonded with data items for interactive privacy analysis preservation at the front-end.

- We implement the prototype of TPA and evaluate its effectiveness with two use cases from the insurance and medical domain, respectively. A qualitative interview points out the pros and cons of TPA from the perspective of domain experts.

## 2 RELATED WORK

In this section, we provide the background of the privacy preserving and review the literature on visualization.

### 2.1 Privacy Preserving Techniques

Data providers will make data sanitization before making it public. There are three dominant technologies:

**Anonymity method**. The most widely used technique for dealing with linking attacks is k-anonymity [36], which is one of the most representative methods. K-anonymity calls all records with the same quasi-identifier an equivalence class. It requires each equivalence class has at least $k$ records. The k-anonymity avoids attackers to identify users by quasi-identifiers with a confidence level no more than $\frac{1}{k}$. However, it cannot prevent homogeneity attacks. For example, the sensitive attributes in a equivalence classes are identical, and the attackers can still confirm their sensitive information. Hence, l-diversity was proposed [25]. If a sensitive attribute of an equivalence class has at least $l$ well-represented count, then the equivalence class is said to be l-diversity. Similarly, if all the equivalence classes meet l-diversity, the dataset can be considered to meet l-diversity. If the distance between the distribution of the sensitive attribute in the equivalence class and the distribution of the sensitive attribute in the whole dataset does not exceed the threshold t, it is considered to meet t-closeness [22]. Unlike the first two methods, it considers the overall distribution of data rather than the specific count, which can balance privacy preserving and data utility. In addition, there are many other variants based on these three methods [20, 35]. But anonymity methods are parameter sensitive, and apply to specific constraints.

**Differential Privacy**. Differential privacy [10, 11, 27] is widely used which has no disadvantages of anonymous methods (only applicable to attackers with specific background knowledge). If the absence of a data item does not significantly affect the output result, the function conforms differential privacy definition. For example, if there is a function that queries 100 items in a certain way and results in the same results as queries for the 99 items, there is no way for an attacker to find information about the 100th item. Therefore, the core idea of differential privacy is that there is only one record for the difference between two data sets, making the probability of the result is almost the same.

**Synthetic Data**. The intuitive advantage of synthetic data is that it is 'artificial data', so synthetic data does not contain real information. Synthesizing data is also presented to protect publishing data from traditional attacks [1, 29, 42]. Therefore, many studies [2, 4, 33] work on similarity between real and synthetic records to measure privacy leakage in synthetic datasets. True, these techniques avoid exposing real data, but as Stadler argues [33], these studies seriously overestimate the ability of privacy preserving. They can't always prevent attacks. Synthetic data is far from the holy grail of privacy preserving data publishing.

### 2.2 Privacy Visualization

Privacy preserving is a part of data processing. Visualization plays a key role in data analysis and processing. Recent literature proves that visualization is gaining momentum in the domain of privacy preserving. Much work has assimilated and expanded the concept of privacy and data mining, analyzed how to reduce privacy leaks, maintain utility, and provide the preserving pipeline.

**Visualization in data analysis**. Data analysis [23, 40] mainly analyzes the relationship between samples from the perspective of distribution, correlation and clustering. Many visualization tools, such as Hierarchical Cluster Explorer [32], PermutMatrix [5] and Clustergrammer [14], are used to analyze the relationship between samples.

Elmqvist and Fekete propose several aggregation designs that aggregate data and convey information about the underlying data [13]. Aggregation can better reflect the statistical information of the data and hide the visual interference caused by individual differences. Aggregation visualization techniques are used more on parallel coordinates, analyzing the utility of a cluster, using summaries of histogram statistics [18].

Tabular is the main way to represent the binary relationship of data. Tabular visualization [15, 16, 30] is extremely scalable because cells in table can be divided into many pixels to show more information about data. Taggle has became one of the most beloved tool [15], which is an item-centric (cells in the sheet) visualization technique that provides a seamless combination of details through data-driven aggregation.

**Visualization for privacy analysis**. In recent years, a growing number of studies focus on visualizing-specific contributions for privacy preserving. Chou et al. proposed a visualization tool to help avoid privacy risks in vision, and designed a visual method based on anonymity technology for social network graphs [6] and time series [7]. GraphProtector [38] guides users to protect privacy by using graphical interactive visualizations to the connection of sensitive and non-sensitive nodes and observe structural changes in the graph of utility.

There are also some studies [3] that analyze how existing visualizations of privacy preserving affect data and how effective they

are. Dasgupta et al. analyze the disclosure risks associated with vulnerabilities in privacy-preserving parallel coordinates and scatter plots [8], and present a case study to demonstrate the effectiveness of the model. Zhang et al. investigate visual analysis of differential privacy [43]. They analyze effectiveness of task-based visualization techniques and a dichotomy model is proposed to define and measure the success of tasks.

**Visualization for privacy preserving**. The preserving pipeline is designed to provide users with a complete processing framework from analysis to protection. Xiao et al. proposed a visualization tool named VISEE [41] to help protect privacy in the case of sensor data sharing. VISEE makes a trade-off between utility and privacy by visualizing the degree of mutual information between different pairs of variables. Overlook [37] was developed for differential privacy preserving of big data. It allows data analysts and data administrators to explore noised data in the face of acceptable delays, while ensuring query accuracy comparable to other synopsis-based systems. Wang et al. [39] propose UPD-Matrix (utility preservation degree matrix) and PER-Tree (privacy exposure risk tree) and developed a visualization tool for multi-attribute tabular based on them. It provides a five-step pipeline of user interaction and iterative processing of data.

These visualization tools are designed to help users troubleshoot potential risks. However, most of them are based on a single exploration domain or automatic algorithms. We find that different schema has great difference in risk analysis and handling (§ 3.2). Our approach allows users to explore the tabular data of different attribute order more flexibly and support different granularity of privacy preserving operations.

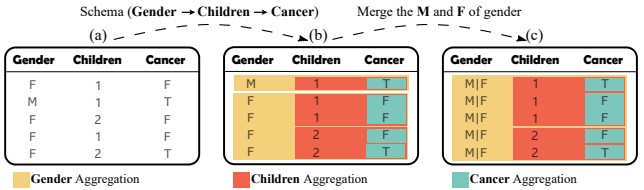

Figure 2: Aggregation for privacy analysis under a schema example. (a) Original sheet. (b) Reordered records by schema. (c) Privacy-enhanced sheet based on merging attribute values.

## 3 PRELIMINARIES, MOTIVATIONS, AND REQUIREMENTS

For ease of exposition, we first denote the relevant entities: 1) **Individuals** whose information are recorded in a sheet. One individual can corresponds to several items. 2) **Data holders**[1] (e.g., institutions, administrators) that own, maintain, and release the sheet.

### 3.1 Preliminaries on Tabular Privacy

The basic privacy risk of sharing tabular data is that individual identity is correlated and may be revealed by the other seemingly harmless attributes. We denote such a unique combination of attributes a **quasi-identifier** for individual privacy. We first give some key definitions in this privacy context.

*(Definition 1)* **Equivalent Class**: A subset of items with equal values on all the focused attributes.

For example, in Fig. 2, if 'Gender' is the focused attribute, then all the items with value 'M' form an equivalent class, while all items with 'F' form another class. According to K-anonymity argument, if the size of a equivalent class is smaller than $k$, then the inside items form quasi-identifiers that identify their owners' identities.

Given a tabular set, a general privacy analysis process (e.g., [39]) involves calculating the size of equivalent classes under additive

---

[1]We use data holder and user interchangeably.

attribute prefix. For example, measuring the size of equivalent class of prefix 'Gender=F' and raising a privacy risk if the number is smaller than $k$; then checking prefix 'Gender=F, Children = 1', et al.

*(Definition 2)* **Schema**: The order of attributes assigned for measuring the size of equivalent class to find a privacy breach.

We denote the process of finding equivalent classes according to the schema as **aggregation**, so that **privacy investigation turns into aggregation in the given order of attributes**. Fig. 2 shows the aggregation under a specific schema (from the left column to the right ones). Attribute of the left-most column (*Gender*) is first investigated by measuring the equivalent class of Male ('M') and Female ('F') separately. Wherein, the 'M' class would breach K-anonymity if $k > 1$, which can be mitigated by merging values 'M' and 'F' together to loose the quasi-identifier ($k = 5$). Then the next dimension (*Children*) is measured by counting items under prefix 'M, 1', 'F, 1', and 'F, 2' separately.

### 3.2 Motivations: Self-defined and Dynamic Schema

Previous studies use a fixed schema during privacy analysis for that analyzing equivalent class for a tabular set is computation complex, especially when there are large amounts of data items with many attributes. As a result, they usually perform analysis according to the original attribute order in the sheet. However, we find that:

**Remark:** *Different schema will yield different privacy risk situations, facilitate distinguished privacy-preserving granularity, and introduce distinct risk handling overhead.*

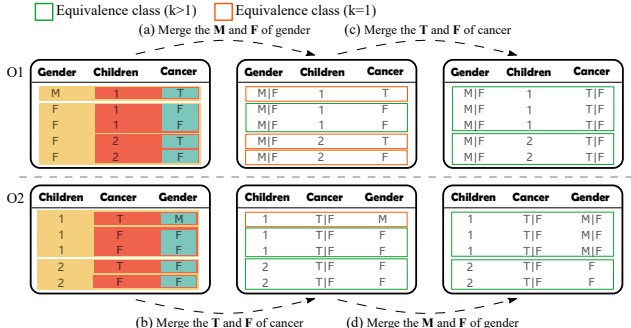

Figure 3: An example of performing equivalent class merging for privacy enhancement under different schemas (O1 and O2). Obviously, the yielded sheets are different.

Taking the merging operation as an example, different schemas correspond to different aggregations and result in different equivalence classes after merging. In the case of Fig. 3, O1 and O2 are the same sheet with different schema. When checking the first attribute, O1 will merge the 'Gender' into $M|F$ (operation a), whereas 'Children' in O2 satisfies $k = 2$-anonymity and will be retained. Then, O1 continues to check 'Children' without merging operation and merge 'Cancer' under the prefix of 'Gender-Children'. O2, on the other hand, merges the 'Cancer' and 'Gender' attributes under the prefix of 'Children'. Finally, O1 has 10 values altered, while O2 got eight changed during aggregation. That is, the schema has a explicit impact on the privacy situations and enhancement level.

In particular, we note that the latter attribute in the schema order has a higher chance of breaching privacy as the equivalent class of a longer prefix (finer granularity) gradually gets smaller, namely, easier to go below the constraint $k$. As a result, the latter attribute may be heavily merged for privacy preserving, losing more utility. Considering this, we point out that **the schema should be assigned by the data holder according to their privacy/utility preference**, e.g., subjectively retaining information of some attributes by putting them in the front. Furthermore, as we will show in § 6, **data holders will dynamically adjust the schema to analyze the risky attributes in**

**coarser granularity** for flexible merging operations. For example, instead of studying many equivalent classes for a latter attribute, one can move it to the front to perform merging on fewer classes.

Yet, existing visualization cannot meet the above dynamic schema intentions, as the change in schema requires a new round of aggregation, which will cause significant latency in online interaction. We are thus motivated to design a new privacy visualization tool that supports schema dynamics.

### 3.3 Requirement Analysis

Through meetings with domain experts, we acknowledge that they are familiar with common privacy enhancement technologies, such as k-anonymity and differential privacy. In fact, they have actively applied these techniques to mitigate risk before data released. On this basis, we discussed the insufficiency in current privacy practice and have identified four main requirements:

**R1: Ability to control schema.** As indicated above, different order of attributes shows different preference on attributes and has different granularity when applying privacy preserving operations. Flexible support for adjustable schema is widely required.

**R2: Multidimensional data analysis.** Users' prior knowledge is important in risk analysis. Even professional data analysts cannot find relation between attributes by simply glancing at a sheet. Heuristic algorithms for risk assessing do not know the semantic knowledge and their results are unreliable. Therefore, domain experts believe that a sketch view on for exploring attribute relations is beneficial.

**R3: Intuitive risk cue.** Prior privacy preserving studies have addressed visual designs for privacy risk. In these realizations, users are reminded that there are risks somewhere without mapping directly to the specific records on the sheet. Thus, it is expected to provide an integrated process for risk presentation and mitigation.

**R4: Operation-granularity utility evaluation.** The sanitization of data inevitably discard some information details. It would blur the data, deviate the statistics, and reduce releasing utility. In particular, as different schema and sanitization operations lead to different utilities, users generally want to attain the utility outputs of the current settings for further involvement.

## 4 BACK-END ENGINE

This section introduces the key techniques of TPA and how they are used in the system.

### 4.1 Data Structure for Order-dynamic Schema

Section 3 discusses the necessity and challenges of adjusting schema dynamically. We propose to adapt data cube as the basis for data management, which has efforts in addressing **R1** and **R2**. With data cube, we design the operation tree and facilitate users to change schema and perform operations with almost no perceptible latency.

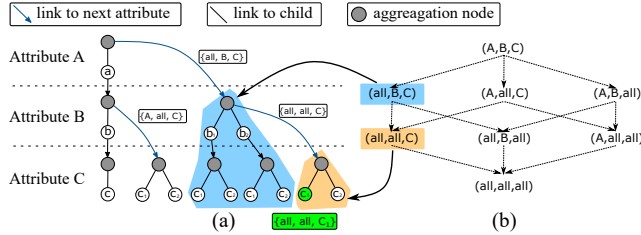

Figure 4: Data cube of a dataset with attributes $D = A, B, C$. (a) Tree-based data cube. (b) Aggregation relationships of the data cube.

**Data cube.** Data Cube [17] is well suited to handle online analytical processing queries. It is a data processing form for statistics of data, such as SUM, MIN, MAX and AVERAGE. Queries have low latencies by pre-aggregated. We introduce the data cube into TPA, and pre-calculated aggregations are used to reduce query latency.

As shown in Fig. 4, (a) shows a tree-based data cube. Similar to a search tree, records are added as a node based on the value of each dimension. However, the data cube also stores the aggregated results (i.e., the subtree pointed to by the blue arrows). The aggregation results store records when a dimension value was ignored. When a query does not care about the values of specific dimensions, data cube can response quickly by access aggregation. For example, The user wants to find all records which attribute $C$ is $c_1$ and doesn't care about other attributes. The result can be obtained by accessing the aggregation of $\{all, all, c_1\}$, which is the green node in the figure.

We did not calculate statistics of records, but store records' indexes into aggregations. TPA use the Nanocubes [24], which is an implementation of the data cube. Nanocubes proposes shared links to avoid duplicate aggregations, thus using less memory. TPA stores categorical attribute and numeric attribute in different ways. For the categorical attribute, creating branches directly based on their values. For the numeric attribute, will have a default branch to store the all original data and others branches have a specific split range. Data cube is built on the server side, which can quickly access aggregations according to the schema.

**Operation tree.** We propose the operation tree for interaction and visualization on the client side. When the user specifies a schema, the operation tree is generated quickly based on the data cube. The operation tree is similar to the data cube in that each node denotes an aggregation and stores indexes of all records in the aggregation. The nodes in the operation tree have the same aggregated order as the schema, only dimensions involved in the schema are stored, which is a lightweight tree designed for front-end (client) interaction. All operations from TPA are performed in the operation tree, such as merging, noise adding, fake data adding and so on. Besides, the node stores privacy-related parameters for visualization (e.g., number of equivalence classes, whether the noise is added, and so on).

When requesting a new schema, TPA will quickly create the operation tree by accessing the aggregations from the data cube. Fig. 5 (a) abd (b) illustrates the process of creating an operation tree, when a user is interested in two attributes and given an schema of $Cancer \rightarrow Gender$. Taking advantages of the data cube, TPA no need to walk through all records to build the operation tree. TPA can quickly find out records of each node (operation tree) just by looking at aggregations of the data cube. TPA is able to create the operation tree with little overhead, even if the user frequently reorders the dimensions.

When we apply certain preserving operations, they are directly performed to the operation tree. Fig. 5 (c) show how the operation tree updated after a merging operation performed. Updating may add or delete nodes and branches, and update the values of node records. Since these changes only tweak the tree structure, they do not incur much computing overhead on the client side, while changes to the operation tree are synchronized to the data cube of the server side, again with no additional delay in interaction.

### 4.2 Utility Quantification

Utility is a summary term describing the value of a given data release as an analytical resource [12], which is essentially a measure of the information obtained from the dataset. There is no accepted measure of utility and few studies focus on utility quantification of tabular data. According to the definition of utility, we consider using **distance** and **distribution** to measure the utility loss. For any values $f_a(x)$ in original dataset (where $f_a(x)$ is the value of the attribute $a$) and handling values $f'_a(x)$ which is obtained after privacy preserving, we use $L_{distance}(F_a, F'_a)$ and $L_{distribution}(F_a, F'_a)$ to denote the utility loss according to the difference in distance and distribution between them. We propose different algorithms to calculate utility losses for numerical data and categorical data, respectively.

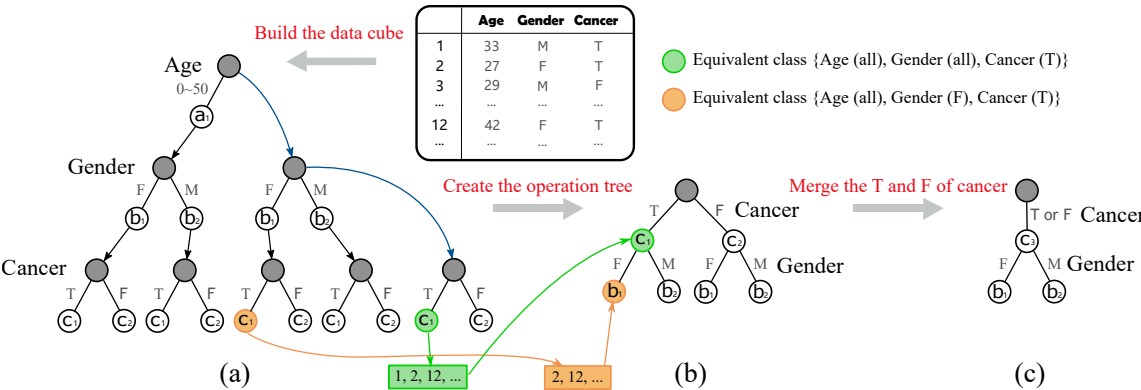

Figure 5: An illustration of creating and updating the operation tree. (a) Data cube is built at the back-end based on the tabular data above. (b) Operation tree is generated based on the data cube. (c) An example of updating the operation tree.

**Numerical Distance**. Inspired by EMD [31], Earth Mover's Distance is used to compare the distance between two datasets. Sort all records of two datasets, and calculate the distance of the corresponding records:

$$L_{distance}(F_a, F_a') = \sum_{i=1}^{n} \frac{|i - j|}{n}, \qquad (1)$$

where $i$ and $j$ refer to the sorted index of $f_a(x)$ and $f_a'(x)$ and $n$ is the number of records.

**Categorical Distance**. Since the Categorical data may be fuzzy, the value of $f_a(x)$ is actually a set. For example, $f_{gender} = \{male, female\}$ represents an uncertain value that the gender of this record may be male or female. First, we calculate $I$ of these two fuzzy sets, where $I$ denotes the number of individual values that can only be taken from one of the sets. Taking $\{a, b, c\}$ and $\{b, c, d\}$ as an example, $a$ and $d$ are individual values, hence the $I$ is 2. Then the distance between sets can be calculated by:

$$L_{distance}(F_a, F_a') = \sum_{i=1}^{n} \frac{2I}{|f_a(x)| + |f_a'(x)|}, \qquad (2)$$

where $|f_a(x)|$ refers to the size of the set (i.e., the number of fuzzy values contained).

**Numerical Distribution**. As a nonparametric test method, K-S test [26] is applicable to compare the distribution of two datasets when the distribution is unknown. We use the K-S test to measure the distribution of numerical distribution and use the p-value to represent the utility loss:

$$L_{distribution}(F_a, F_a') = 1 - p. \qquad (3)$$

**Categorical Distribution**. To measure the distribution of fuzzy sets, we first get the global distribution of all possible values. For an attribute $a$, count the number of occurrences of all values $C = \{c_{a_1}, c_{a_2}, \ldots, c_{a_n}\}$, where $c_{a_n}$ refers to the number of values with $a_n$. Given a fuzzy set $f_a(x)$, counting each possible value $a_n$ by $c_{a_n} = c_{a_n} + \frac{1}{|f_a(x)|}$. After getting the global count, the distance of each value can be calculated by:

$$L_{distribution}(F_a, F_a') = \sum_{i=1}^{n} \frac{|C - C'|}{n}. \qquad (4)$$

## 5 FRONT-END VISUALIZATION

As shown in Fig. 6, the front-end of TPA works in 5 steps: importing, building data cube, privacy analysis, privacy preserving, and exporting. Among them, (c) and (d) are of the most concern for data holders. Being at the heart of visualization and interaction, these two steps work iteratively by presenting risks and performing enhancement until privacy and utility are both satisfactory.

### 5.1 Importing

As the first step in the pipeline, the user needs to upload the data sheet here. TPA will attempt to automatically identify the attributes type (categorical or numeric), and user can correct possible misjudgments by manually setting the type. Once the attribute type is determined, it cannot be modified in subsequent steps.

### 5.2 Building Data Cube

After receiving the sheet uploaded at the first step, TPA will build the data cube for management and create a session. The session created is used to respond to requests for schemas and to keep track of updates to the operation tree.

### 5.3 Privacy Analysis

Fig. 1 (a) shows how the schema is modified. This widget has two boxes (left and right), and user changes the order by moving the attributes in these two boxes. The first time got to this step all the attributes are in the right side area, and users can select the interested attributes and move them to the left. Users can also add or remove interested attributes at any time. The attributes of the left box can be dragged at will to adjust the aggregated order. Thanks to data cube applied, any changes to the schema will instantly generate the corresponding operation tree. In addition, clicking an attribute can mark it as sensitive (used for l-diversity and t-closeness).

**Abstract**. Aggregations have sorted records in equivalent classes according to the schema, but dozens or even hundreds of lines of records are hardly to be summarized. To help data holders understand and analyze the relation between attributes (**R2**), we design the visual abstract. As shown in Fig. 1 (b), TPA provides a global abstract, which shows the distribution and proportion of values. In addition to the global summary, TPA supports draw abstract for any aggregation selected. Clicking on the left of the record to collapse or expand the aggregation, and draw abstract for the collapse one. There are two types of the abstracts, as shown in Fig. 7:

- The categorical abstract in (a). Its value distribution is represented by the percentage of color block. For fuzzy values, such as null values and uncertain values, are bisected among all possible blocks. The light (upper) part of the color block

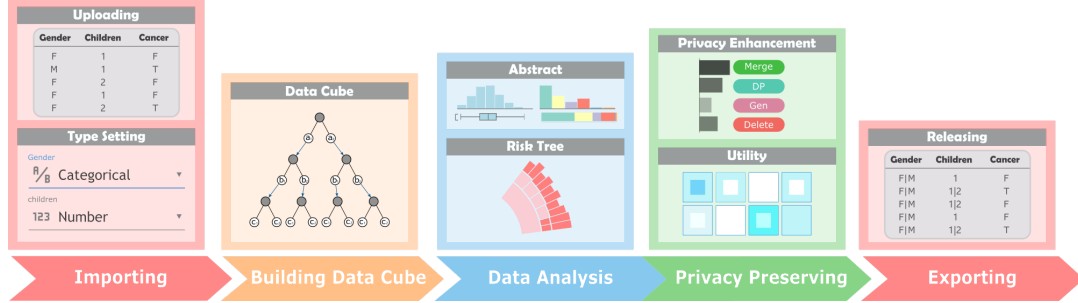

Figure 6: TPA visualization framework, a 5-step pipeline: import the data sheet, build the data cube, iterate to analyze and deal with privacy risks, and finally export the data sheet.

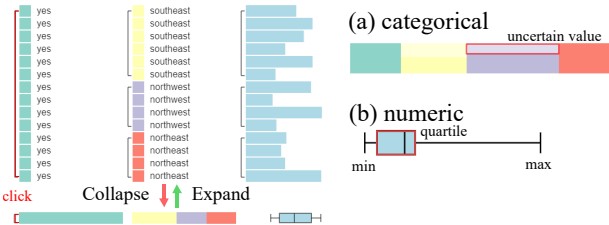

Figure 7: An abstract design for focused data items (e.g., equivalent class) summarizing. An example abstract of the categorical attributes (a) and numeric attributes (b).

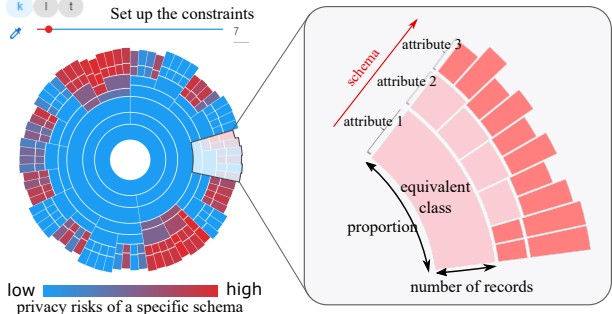

Figure 8: A risk tree design for intuitive perception of privacy risks in aggregations.

refers to the uncertain value. By observing the proportion of the light part, users can know how many records apply the merging operation.

- The numeric abstract in (b), based on a box-plot design. The box-plot clearly shows the extreme, quartile, and mean values of the aggregation.

With summaries by the abstract, users can quickly get the information of the selected aggregation and the relation between the data of different attributes, which is helpful for data analysis.

**Privacy risk tree**. Abstract can guide data analysis, and help to explore data relations, but users also like to tell them directly where the privacy risks are (**R3**). We came up with a more intuitive visual design, the risk tree, locating privacy risks according to anonymity technologies. Fig. 8 illustrates the risk tree widget. A selector on the top left of the widget allow user to select a specific anonymity technology from k-anonymity, l-diversity and t-closeness. The constraint parameters are set by the slider. Risk Tree consists of layers of pie charts, with the layers from inside to outside corresponding to the given schema. The division of piece at each layer represents the distribution of the value of this attribute, and each piece is a corresponding node (aggregation) from the operation tree. Calculate whether each piece satisfies the constraint based on the parameters set by the user, and map the privacy risk of aggregations to different colors. When the block does not meet the constraint, the color is calculated by linear interpolation, and the color can relay the degree of risk of each aggregation. Users can hover to view specific aggregation information, and click to quickly jump to it's location in the main view.

Due to the different granularity of each layer, the actual priority of risks are different. Obviously, the aggregation node of the outer layer has fewer records, exposing fine-grained privacy risk easily. On the contrary, the high risk color of the inner aggregation indicates that the aggregation has large-scale leakage and should be paid more attention to. A simple understanding is that the risk of inner

aggregations means that an attacker can use less information to identify items and should be dealt with first.

### 5.4 Privacy Preservation

The privacy risks identified in the previous step could be addressed in the this step. TPA provides four operations for privacy enhancement: merging, noise injection, fake data injection, and removing. Operations other than merging require a selection of records. As shown in Fig. 9 (b), user can select records by ctrl+clicking equivalent classes or records.

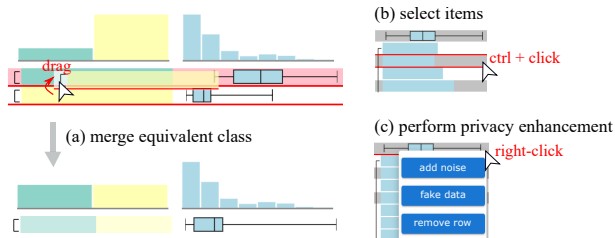

Figure 9: Preserving operations. (a) Merging operation. (b) Select records. (c) Open operations menu.

**Merging** . Merging is the primary preserving operation, which prevents an attacker from identifying items by making the value fuzzy. Two equivalent classes can be merged when all prior attributes of them have same values (i.e., the two nodes in the operation tree have the same parent node). As shown in Fig. 9(a), dragging one folded class onto another to merge them. The value of these two classes will be updated from a concrete value (**A**, **B**) to an uncertain value (**A|B**). Besides, the two aggregations that are merged will exist

as a new class in the operation tree, so user can continue to merge it with other classes which have the same parent.

**Adding noise**. The noise operation applies to numeric attributes. By clicking the 'Add noise' of the menu, a new view for noise operation is shown in Fig. 10. The view shows the histogram for all numeric attributes, and the number of bars in the histogram determines the granularity of bins, which can be set by tweaking the slider above. The noise operation adds Laplace noise to the data based on differential privacy. One can click the switch in the upper right corner to set the noise parameter, and drag the white dot in Fig. 9(b) to set $\lambda$ of Laplace for each bin that how much noise to add. After parameters are set, view will prompt some red lines which denote the fluctuations of each bin after adding noise.

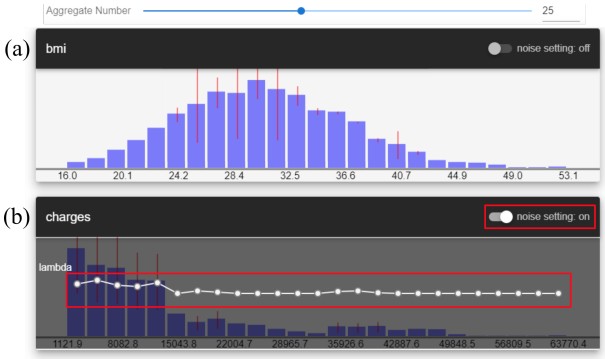

Figure 10: A visual design for adding noise.

It is unreasonable to add noise to all data indiscriminately. As shown in Fig. 11, TPA provides a matrix view for data analysis and filter data of interest.It shows the two-dimensional distribution of attributes, where the x-axis of the chart is the attribute above the view and the y-axis is the corresponding numeric attribute. A Scatter-plot is used for numeric-numeric combinations and a grouping Box-plot is used for numeric-categorical combinations. User can select data by brushing and clicking, at which point the noise operation will only be applied to the selected data.

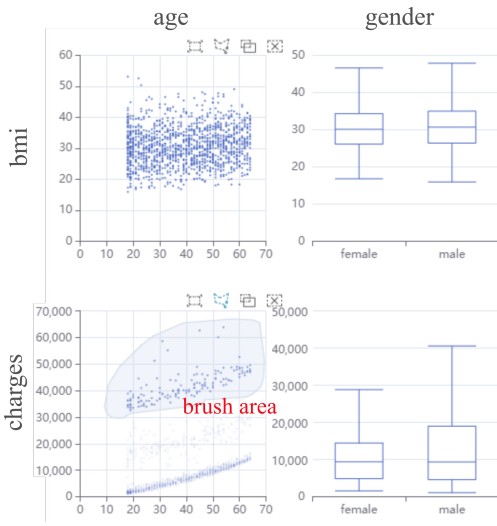

Figure 11: The matrix view presents the two-dimensional distribution of data and provides data filters.

**Adding fake data**. The fake operation uses CTGAN [42] to generate synthetic records and adds these records into sheet to confuse attackers. After clicking 'Generate fake data' in the menu, TPA will use the selected records as training inputs to generate synthetic records. Synthetic data is not always effective in preventing leakages, but it provides a method that does not require other prior knowledge. Since the synthetic data have similar distribution as the training inputs, the utility loss can be controlled to some extent.

**Removing records**. Sometimes, users want to remove records directly (e.g., outlier data). The removing operation can be applied to remove the selected records from the sheet.

**History View**. The history view records all privacy enhancement operations applied. As shown in Fig. 12, the view lists historical states and their detail, allowing user to go back to the historical state. It also provides the user the number of records that are affected. This helps users understand the granularity of each operation. In addition, users can compare utility losses by selecting two historical states.

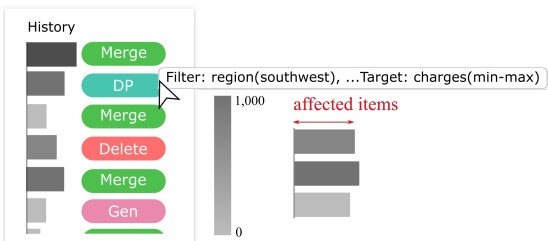

Figure 12: The history view records the historical states.

**Utility analysis.** For any preserving operation, whether it is modifying the original value or adding/removing records, will result in a loss of utility. Thus, user want to see how utility changes with each operation (**R4**). TPA uses the measure introduced in section 5.2 to estimate the utility loss by calculating the distance and distribution. To compare utility changes in each operation, we propose the utility comparison view (Fig. 13). Users can select two historical states at history view to compare. When we select a historical state, TPA will compute the result from applying the first operation to the operation selected, and then calculates the difference in utility between selected state and the original sheet.

To compare two different states, we utilize a **superimposed matrix** to visualize the changes in two historical states. The rows represent algorithms to be compared and the columns represent attributes. Each cell is divided into an outer region and an inner region, with the background color saturation representing the difference of the utility in two different state. The higher the saturation, the more the differs from the original data in this attribute (high utility loss). The view is designed to help users understand changes in utility.

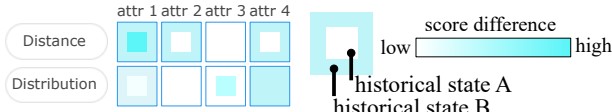

Figure 13: Comparison of two historical states, indicating the difference in utility loss.

## 5.5 Exporting

The analysis and preserving loop stops if data holder considers the privacy and utility situation are satisfactory. The corresponding sheet is such downloaded and released.

## 6 CASE STUDIES

We conduct two case studies with the prototype of TPA, with data from the insurance domain and medical domain.

## 6.1 Analyzing the Medical Cost Dataset

The medical cost dataset is an insurance-billed personal medical cost obtained from a book [19]. It has a sheet which shows the age, gender, bmi, children (number of children), smoker, region and charges of 1,339 personal information. This dataset has been sanitized before releasing. In this example, we assume that the dataset collects data from the same hospital and the attacker is most likely to identify individuals through linking attacks.

The records of children are numerical. Obviously, the number of children doesn't vary that much and people focus more on whether the patient has a child. Therefore, we mark children as categorical in step 1. After completing the basic setup, we continue to conduct the preserving pipeline.

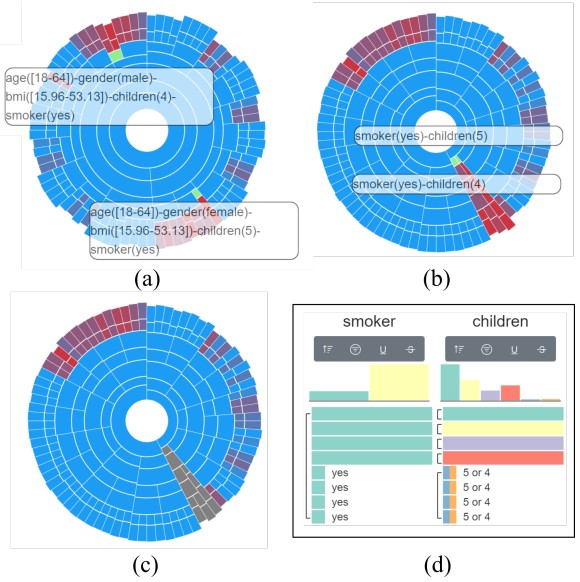

Figure 14: Process of dealing with privacy risks in the Medical Cost Dataset. (a) Privacy risks are associated with smokers. (b) Adjust the schema to identify high risk aggregations. (c) Use the merging operation to address risks. (d) Affected records.

In the case of unfamiliar data, anonymous analysis can be considered first. By setting the k-anonymity constraint of the risk tree as $k = 7$ and observing the visualization in Fig. 14 (a), we find that there are two prominent high risk aggregations. By jumping to the specific aggregations in the sheet, we find they are all smokers. In this case, we can merge 'yes' and 'no' of the attribute "smoker", but many non-smoking data will also be blurred. Comparing aggregations of the previous attribute 'children', we find that both of them have more than four children. It's easy to understand that people with more children are the minority that are easier to identify. Thus, we adjust 'smoker' and 'children' to the front of the order. At this schema in Fig. 14 (b), the new risk-tree indicates their high risk aggregations. As a result, the patients with high risk are those who have more than four children. As shown in Fig. 14 (c) and (d), by merging aggregations of smokers who have '4' and '5' children, risks have been reduced, and only four records related to risks are modified.

When we adjust the 'charges' and 'smoker' to the front of the order and collapse the aggregation of 'smoker' (Fig. 15), abstracts in (a) show an interesting pattern that smokers have much higher charges than non-smokers. This pattern indicates that attacker can simply predict their charges by whether patients smoke or not, with a high degree of confidence. To prevent potential background knowledge attacks, we focus on smokers to protect their privacy, since

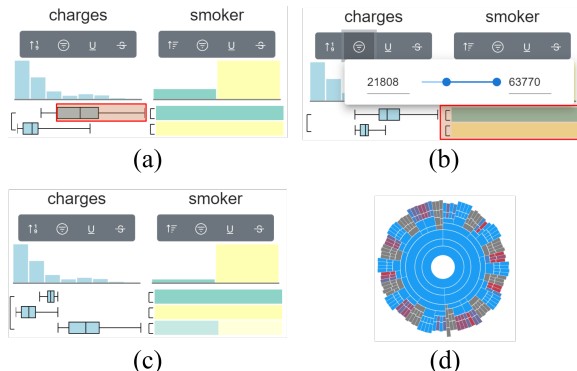

Figure 15: Identify potential risks through abstracts. (a) Smokers have extremely high charges. (b) Filter the records to be protected by a filter. (c) and (d) Result of applying preserving operations.

smokers are a small group. Therefore, we set a filter to find out high charges of both non-smokers and smokers, and merge aggregations of them in (b). (c) indicates that we make high charges records fuzzy which protect the privacy of smokers. Besides, it is reasonable to keep the low charges data which are mostly non-smokers (the majority of people).

## 6.2 Analyzing Personal Key Indicators of Heart Disease

This dataset comes from the CDC (Centers for Disease Control and Prevention) [28], which collects data on the health of U.S. residents. Each record has 300 attributes, including various indicators of the body. According to a CDC report, heart disease is the leading cause of death in the United States. Considering indicators related to heart disease, we narrowed it down to 12 attributes and randomly selected 20,000 records for this example.

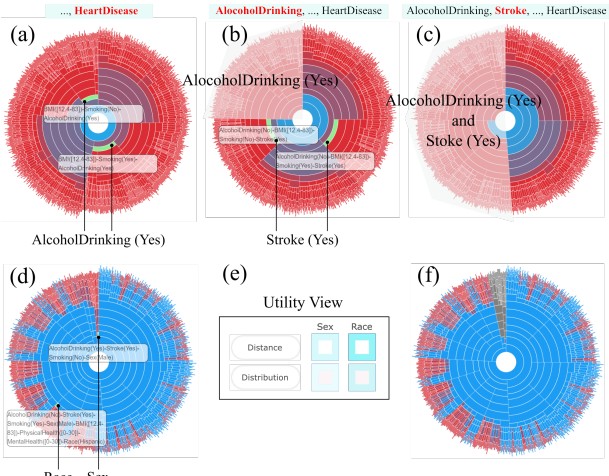

Figure 16: For a hight-dimensional complex dataset, t-clossness is used to explore dimensional correlations and locate high risk aggregations. (a), (b) and (c) Iterate through the schema to find high correlated attributes. (d) and (e) Compare the utility loss after using the preserving operation on 'Race' and 'Sex'. (f) Result of applying preserving operations.

Patients certainly don't want to expose their disease. In this example, we focus on analyzing and dealing with privacy risks related to 'HeartDisease' attribute. From the perspective of publishers, we should first find out what other attributes are related to the disease. We set the 'HeartDisease' as a sensitive attribute and adjust it to

the end of the order. As a result in Fig. 16 (a), the t-closeness view in risk-tree points out that the distribution of heart disease among drinkers was clearly different from the global distribution. It can be considered that drinking is highly correlated with heart disease. We move 'AlocholDrinking' to the front of the order and look at the risk-tree again. The new view (b) shows that "Stoke" also has a significant effect on the distribution. Thus, we move 'Stroke' after 'AlocholDrinking'.

We have moved high correlated attributes to the front of the order, and adjusted schema is easier to locate risks than a random schema (c). After switching to the K-anonymity view, we find some aggregations with salient high risk in branches of the 'Sex' and 'Race'. To reduce risk, we merge aggregations of 'Sex', which is shown in Fig. 16(d). Jump to the high risk aggregation and try to deal with two attributes separately by merging operation. Fig. 16 (e) indicates the comparison of the feedback from utility view, we find that to merge 'Sex' has less utility loss than to merge 'Race'. Therefore, merging aggregations of 'Sex' is a better choice to reduce risks.

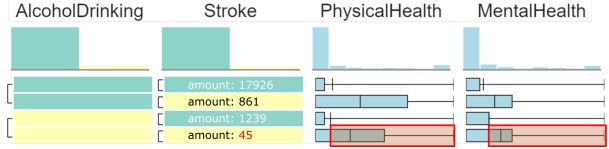

Figure 17: The result of abstracts indicates that people with stroke have more physical and mental health problems.

To further explore the risks, we collapse the attribute 'Stroke' (Fig. 17). Abstracts show that people who have had a stroke tend to have high value of mental and physical problems. The proportion of people who had both stroke and alcohol drinking is small, and stroke is highly associated with heart disease. Although health scores are less sensitive. That also means health scores are also more likely to be collected by attackers, which should be blurred for patients with heart disease. As shown in Fig. 18, We filtered and selected stroke and alcohol drinking among patients with heart disease, adding noise to the high values of mental and physical problems.

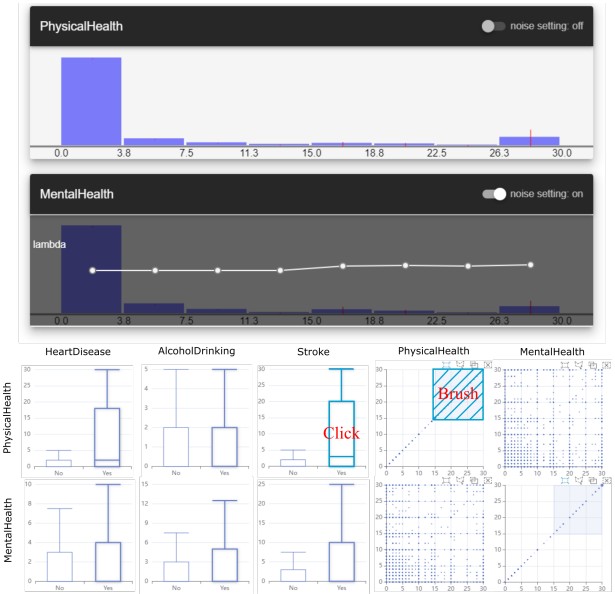

Figure 18: Add noise to blur the health score and protect the privacy of people who smoke, stroke and have heart disease.

For a dataset with 12 dimensions and 20,000 records, taking a long time to calculate once aggregations. Taking advantage of the data cube, even such a high-dimensional dataset can still interact in real-time and dynamically adjust the aggregated order.

## 7 QUALITATIVE DISCUSSIONS

We conducted interview with four domain experts on the applicability of TPA in real-world scenarios. These users are experienced in data analysis and often work with tabular data. They commented positively on our work and indicated suggestions for improvement.

### 7.1 Effectiveness

Interviewees agreed that TPA was effective in data analysis, especially in aggregation abstract that help them to grasp the value distribution of attributes and the correlation between attributes in the dataset (**R2**). They favored the function that they could adjust the schema in real-time (**R1**), and also appreciated TPA's capability to efficiently handle big datasets. One of the users said that *it was difficult to effectively analyze the risks of data sets in the past when faced with high-dimensional data sets*. When used in conjunction with risk tree, dynamic adjustment order were considered to help perceive privacy risks intuitively (**R3**). In addition, TPA saved them a lot of time than other visualization tools, by providing more preserving operations and allowing them to control the granularity of them (**R4**).

### 7.2 Limitations

However, some users pointed out that the interaction design of the prototype was not good enough, even though we instructed users how to use TPA in prior. Further, some supposed that the utility view may be of limited use. While the utility view could remind them of the differences between the current state and the original one, they still don't understand how those differences mean. Some users also suggested providing a recommendation scheme function to help to carry out privacy enhancement operations. This indicates that, whereas TPA is designed to provide users with high flexibility, they can often get lost in the choices, thus providing some recommended actions shall be a good way to get started quickly.

### 7.3 Future Work

Considering that data will be shared to work for specific analysis tasks, we plan to extract patterns for those tasks (e.g., extreme values of samples, clustering, etc.). By indicating the pattern differences before and after privacy preserving, one can more easily take balance between privacy and utility. We will also improve the interface and provide support for more diverse data type, like time, location, sequence, etc.

## 8 CONCLUSION

We propose a visual tool, TPA, for privacy protection of tabular data. Our design helps users analyze multidimensional data relationships and identify potential privacy issues. In addition, we provide users with some preserving operations to reduce privacy risks and a utility view is designed to help control the utility loss of operations. By introducing data cube, we have implemented a system that support user exploring any aggregated order in real-time, allowing users to analyze privacy risks from different perspectives and flexibly control the granularity of preserving operations. We use two real datasets to demonstrate that TPA can handle all kinds of data, including big datasets and high dimensional datasets.

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
