# OpenReview forum: "A Visual Tool for Interactive Privacy Analysis and Preservation on Order-Dynamic Tabular Data"
_graphicsinterface.org/Graphics_Interface/2022/Conference — Submitted to GI 2022_

### Official Review · Reviewer_QvkH · 2022-04-12
**Interesting system, but the evaluation and discussion are lacking.**

**Rating:** 4
**Confidence:** 3

**Review:**

This paper presents TPA, a privacy analysis tool to help users understand relationships between their data and privacy issues. TPA supports four privacy preservation techniques: merging equivalent classes, inserting noise and fake data, and removing data. The discussions of the related work and the system itself are thorough. I believe that the system attempts to address an important problem.

My biggest concern with this paper is the evaluation. It seems very shallow and I personally did not feel like I learned anything after reading it. While the case studies are valuable for showing how one might use the system, the results from the interviews seem superficial. I expect a lot more discussion of the strengths and weaknesses of TPA, especially since it's listed as one of the main contributions in the introduction. For example, what about the interaction design was "not good enough"?

Furthermore, large claims are made without any justification or explanation. For example, the paper notes that that TPA "saved [interviewees] a lot of time than other visualization tools." How much time was saved? How was this measured? Was this through a controlled study? What were the other visualization tools? The discussion in general is lacking and I feel like broader topics should be discussed outside of the interview results (e.g., how TPA could be adopted into existing workflows and analysis tools).

The way the case studies are described, it seems like the user would have to have deeper knowledge of their data, including how it was collected and how it will be used. For example, marking the number of children as categorical because people focus on whether the patient has a child or not does not seem "obvious" to me. It would be nice to see some discussion of how much knowledge is required prior to using TPA versus how much is gained by using it, especially when someone may not know how the data is being used.

Other:
- Several typos and grammatical errors throughout the paper.

---

### Official Review · Reviewer_kNZh · 2022-04-13
**A new technique to finding privacy vulnerabilities and utility in tabular data**

**Rating:** 5
**Confidence:** 2

**Review:**

This paper provides a presentation of the algorithms and techniques implemented in a new tool, TAP, for assessing the vulnerabilities that may increase the ability to identify individuals within a tabular dataset(e.g. medical data). In addition, manipulating dataset to reduce these vulnerability may decrease its utility, which may then reduce the usefulness of any analysis of that data in making decisions/policies/conclusions about the results. They outline four requirements for adequate privacy protection of datasets using analysis and visualization tools, which are used to assess TAP with two case studies and expert reviews.

The background sections seems fairly thorough and provides sufficient background to the issues of privacy compromises of tabular datasets.  However, some of the terminology such as "parameter sensitive" require further explanation.

The examples provided in section 3 are helpful and interesting to gain a better understanding of the scope of the issues. However, it would be useful to have a further explanation of the potential types of compromises of the merged/adjusted data. It would also be helpful if a description of how the potential for a data breach/theft is reduced in these examples.

In section 4, it is unclear as to how missing data are treated.

In Figure 15b, the caption reads "filter records to be protected by a filter," it is unclear what this means. Which filter?

The case studies provide good examples of the data manipulation and output features of TAP can be used. The interface operations seem unclear. For example, how would a user know which data to aggregate/filter. Is this a trial and error process or does TAP provide some default filters?

In section 7.1, it is unclear who the user is. Is it the interviewee?

Finally, this paper is quite poorly written and difficult to read. For example, technical issues include:
- In title, “Interactively” should be "Interactive"
- Figure 1 should not appear first in paper. It should occur after the abstract and introduction, after the first reference to it in the text.
- The last sentence in first paragraph of the introduction is confusing.
- There are many sentences where the verb and object do not match or the sentence is missing something (e.g., “Data providers will make data sanitization before making it public….,” I believe this should be “Data providers sanitize the data before….). This paper is difficult to read as a result.
- Data are plural, use plural verbs instead of singular (e.g., “is” should be “are”)
- Figures should always appear after their reference in the text.
- Page 4 has “abd” instead of “and”
- It’s = it is, it is not a possessive form of "it"
- Avoid using contractions at all in an academic paper, accept when it is a verbatim quote.

---

### Official Review · Reviewer_MAD6 · 2022-04-13
**An interesting and important application area - good case study of the power of visualization**

**Rating:** 9
**Confidence:** 4

**Review:**

This paper presents an application for working with datasets to carry out privacy-preserving modifications while balancing the loss of information that such modifications incur. The system allows for the selective merging of attribute values in order to provide k-anonymity protection and uses visualization of various metrics across the data schema space to assist a data curator/owner to decide which operations to carry out to achieve the k-anonymity with minimal data loss (to important/key attributes).

The paper describes the problem clearly, and outlines the background well. I am not an expert in this type of work, so I found it informative and interesting to read sections 2 and 3 which outline related work and the background knowledge needed to understand the approach. Section 4 introduces the back-end processes for analyzing the privacy risks as well as the information loss costs of operations, using a data cube. Several measures of dataset utility are introduced in order to allow for quantification of utility loss under proposed schema operations. The visualization supports decision-making on these operations, which are carried out efficiently due to the data cube structure. I found the 'abstract design' to make sense as a representation of attribute distributions. Figure 6 clearly describes the process. Where I had more trouble was in interpreting the sunburst diagrams of the schema space. Figure 8 was reasonable as an explanation, but I think it would be helpful to expand this figure and fully label some cells in terms of the schema they represent. However, in the labelling, attributes 1, 2, 3 are actually attributes 5, 6, 7, right? Aren't the inner rings (not shown in the enlargement) also representing attributes? I also was having trouble figuring out where a schema that had no operations (no merges) would be represented here. I think this would be clearer with a deeper explanation and some examples. The challenges I was having with these figures carried through to become issues understanding the processes illustrated in the use cases Figure 14 and 16. I think I got the gist, but the specifics were not clear. I also didn't know where the tooltip overlays on these figures were linked. For example, F14(b) has two tooltips but only one highlighted (green) cell.

My other point of confusion was on when to apply the noise brush. I think the selective addition of noise is a very sensible decision, but in Figure 11 it wasn't clear why the brush was applied as it was. I also didn't understand why the bmi x age chart (top of F11) was shown at all. Is it to show that noise is not needed here? A better explanation or caption is needed to explain how someone would know where to apply the brush.

The qualitative discussions need to be toned down a bit, especially the claim that "TPA saved them a lot of time" since the participants in these discussions were shown a demo and did not actually use TPA. There was no comparator given, so saving time is not informative unless we know what their normal process is.

The paper is generally interesting, a good demonstration of visualization, and shows a fully-realized prototype of a privacy preservation system, grounded in privacy data theory. I think it is a good fit for GI. However, the writing needs work. There are many awkward sentences and disfluencies, as well as typos. I'll list the ones I marked below.

Minor:
- P1 - spell out TPA first time
- P2 - a better definition of k-anonymity is needed when it is first introduced; don't assume the GI audience will be familiar with this term; the forward reference to section 3.2 is not helpful
- P2 - "Elmqvist and Fekete... data [13]." -> this sentence is awkward and vague.
- P2 - "Tabular is the main way to represent the binary relationship of data." -> not sure what this means in terms of binary relations?
- P2 - "Taggle has become one of the most beloved"... this is an odd way to describe a research approach. Also, not sure there is evidence to support this statement (it's a good paper, but has just 20 citations).
- P2 - "There are also some studies [3]" -> if there are "some" there should be multiple citations here.
- P3 - "We find that different schema has great difference is risk analysis and handling." -> rewrite, unclear
- P3 - "can corresponds" -> "can correspond"
- P4 - "TPA no need to walk through" -> grammar
- P4 - Reference to Figure 5(c) is a forward reference and skips 5(a) and 5(b) - generally not good style
- Section 5 intro - references are made to "(c) and (d) are of the most concern" - I figured this means "privacy analysis" and "privacy preserving" but the list was not labeled with letters so it isn't clear.
- P5 - "first time go to this step" -> "first time at this step"
- P6 - "but users also like to tell them" -> unclear. Should this be "but users also like the system to tell them"?
- P6 - "Merging ." (remove space)
- P7 - "user wan to see" -> "the user wants to see"
- P8 and P9 (and Figure 16) - "alcohol" is spelled many different ways, "alocohol", "alochol" etc.
- P9 - ''Stoke -> should be ``Stroke (quote is backwards and word misspelled)
- The teal to white color scale of F13 doesn't offer a lot of luminance contrast

The references need to be proofread, especially for capitalization of journal and conference names, as it is inconsistent.

---

### Decision · Program_Chairs · 2022-04-17

Reject